# A Note on the Connection between Non-Additive Entropy and *h*-Derivative

**DOI:** 10.3390/e25060918

**Published:** 2023-06-09

**Authors:** Jin-Wen Kang, Ke-Ming Shen, Ben-Wei Zhang

**Affiliations:** 1Key Laboratory of Quark & Lepton Physics (MOE), Institute of Particle Physics, Central China Normal University, Wuhan 430079, China; kangjinwen@mails.ccnu.edu.cn; 2School of Science, East China University of Technology, Nanchang 330013, China

**Keywords:** non-additive entropy, *h*-derivative, *S*_*h*,*h*′_-entropy

## Abstract

In order to study as a whole a wide part of entropy measures, we introduce a two-parameter non-extensive entropic form with respect to the *h*-derivative, which generalizes the conventional Newton–Leibniz calculus. This new entropy, $S_{h,h^{\prime }}$, is proved to describe the non-extensive systems and recover several types of well-known non-extensive entropic expressions, such as the Tsallis entropy, the Abe entropy, the Shafee entropy, the Kaniadakis entropy and even the classical Boltzmann–Gibbs one. As a generalized entropy, its corresponding properties are also analyzed.

## 1. Introduction

Since it was proposed over one hundred years ago, the conventional Boltzmann–Gibbs (BG) statistics has been developed very delicately and successfully with wide applications in many disciplines. During the last few decades, however, people noticed that more and more systems are difficult to be described by this simple BG distribution, such as the long-range interactions [1], the gravitational systems [2], the Lévy flights and fractals [3], and so on [4]. In order to cope with this challenge, some attempts have been made to generalize the BG statistics. Among them, the most investigated formalism is the non-extensive entropy. It was inspired by the geometrical theory of multi-fractals and its systematic use of powers of probabilities by C. Tsallis [5]:(1)$$\begin{array}{c} S_{q}=k_{B}\frac{1-\sum_{i=1}^{W}p_{i}^{q}}{q-1}, \end{array}$$
where $k_{B}$ is the Boltzmann constant (hereafter we assume $k_{B}=1$ for simplicity) and *q* stands for the Tsallis non-extensive parameter. It describes the departure of non-extensive statistics from the BG one. This entropy goes back to the usual BG form when q→1. For more than two decades of researches and developments, the Tsallis entropy has been successfully applied to various domains: physics, chemistry, economics, computer science, biosciences, linguistics, and so on [5,6,7,8,9,10]. For the average charged-hadron yields in inelastic non-single-diffractive events, V. Khachatryan et al. observe it as the Tsallis distribution [11,12]
(2)$$\begin{array}{c} E\frac{d^{3}N_{ch}}{dp^{3}}=\frac{1}{2\pi p_{T}}\frac{d^{2}N_{ch}}{d\eta \;dp_{T}}=C\frac{dN_{ch}}{dy}{\left(1+\frac{E_{T}}{nT}\right)}^{-n}, \end{array}$$
where $E\frac{d^{3}N_{ch}}{dp^{3}}$ is for the function of spectra with *E* for the total energy of the particle and *p* for its momentum, η denotes the pseudorapidity with *y* for the rapidity, $p_{T}$ stands for the transverse momentum, *C* is for its normalization constant, *T*, a variational parameter representing the temperature when the system reaches equilibrium, *n* is the fitting parameter which connects with Tsallis’ *q* by n=1/(1−q), $y=0.5\ln\left[\left(E+p_{z}\right)/\left(E-p_{z}\right)\right],E_{T}=\sqrt{m^{2}+p_{T}^{2}}-m$, and *m* is the charged pion mass. The data fitting results show that the Tsallis distribution can well-describe both the low-$p_{T}$ exponential and the high-$p_{T}$ power-law behaviors [11,12]. One application in astrophysics is the study of the distribution of asteroid rotation periods from different regions of the solar system and diameter distributions of near-Earth asteroids (NEAs) [13]. A. S. Betzler and E. P. Borges analyze two samples from different years. They discover that the distribution of diameters of NEAs obeys a Tsallis-like distribution, and the rotation periods of asteroids can be well-approximated by a Tsallis–Gaussian function. According to the first conclusion, there should be 994±30 NEAs with diameters greater than 1km [13]. In another example, Y. Wang and J. Du study the viscosity of light charged particles in weakly ionized plasma with the power-law Tsallis-distributions using the generalized Boltzmann equation of transport and the motion equation of hydrodynamics [14].

The Tsallis entropy is indeed not unique. By now, a lot of different expressions of the non-additive entropies have been proposed, for instance, the Kaniadakis entropy [15], the Shafee entropy [16], the $q-q^{-1}$ symmetric modification of Tsallis entropy [17], and the two-parameter $\left(q,q^{\prime }\right)$-entropy [18]. These expressions were obtained in quite different ways and investigated by distinct motivations. Therefore, it will be of great interest to find the relationship among these formulas or to find a simple formula to study them as a whole.

In this paper, we first introduce a two-parameter non-additive entropy, $S_{h,h^{\prime }}$, based on the *h*-derivative. The *h*-derivative is known as a mathematical generalization of the normal Newton–Leibniz calculus. We address that $S_{h,h^{\prime }}$ unifies different types of expressions of non-extensive entropies; namely, it can connect a family of non-extensive entropies. On the other hand, we also discuss its properties in order to better understand this newly established non-additive entropic function.

## 2. h-Derivative

In the conventional mathematical theory, the Newton–Leibniz derivative is defined as:(3)$$Df\left(x\right)\equiv \frac{df(x)}{dx}=\lim_{\delta \to 0}\frac{f(x+\delta )-f(x)}{\delta }.$$Classically, most of the physical quantities are continuous, and it is natural to apply the Newton–Leibniz derivative. In quantum physics, on the other hand, all the physical quantities will be quantized; people then try to develop quantum calculus, which utilizes the discrete forms of derivatives instead and presents a generalization of this Newton-Leibniz derivative.

One formalism of quantum calculus is the *h*-derivative [19]. For an arbitrary function f(x), its *h*-differential is defined as follows:(4)$$d_{h}f\left(x\right)=f\left(x+h\right)-f\left(x\right).$$It is easily verified that
(5)$$d_{h}x=h,$$
and
(6)$$d_{h}\left(f(x)g(x)\right)=f\left(x+h\right)d_{h}g\left(x\right)+g\left(x\right)d_{h}f\left(x\right).$$Thus, can we obtain the *h*-derivative of f(x):(7)$$D_{h}f\left(x\right)\equiv \frac{d_{h}f\left(x\right)}{d_{h}x}=\frac{f(x+h)-f(x)}{h}.$$When f(x) is differentiable, the following property is obviously obtained:(8)$$\lim_{h\to 0}D_{h}f\left(x\right)=\frac{df(x)}{dx},$$
which is nothing but the definition of the conventional Newton–Leibniz derivative. Note that we need the function f(x) to be continuous for the Newton–Leibniz derivative, but this requirement becomes unnecessary for the *h*-derivative.

Next, some basic rules of this *h*-derivative are listed:Sum and differenceConsidering the sum and difference rules of the *h*-derivative, we have
(9)$$D_{h}\left[f(x)\pm g(x)\right]=D_{h}f\left(x\right)\pm D_{h}g\left(x\right).$$Product and quotient rulesAs for the product and quotient rules,
(10)$$\begin{array}{ccc} D_{h}\left[f(x)g(x)\right] & = & f\left(x\right)D_{h}g\left(x\right)+g\left(x+h\right)D_{h}f\left(x\right), \end{array}$$
(11)$$\begin{array}{ccc} D_{h}\left[\frac{f(x)}{g(x)}\right] & = & \frac{g\left(x\right)D_{h}f\left(x\right)-f\left(x\right)D_{h}g\left(x\right)}{g(x)g(x+h)}. \end{array}$$*h*-derivative of elementary functionsSome other basic calculations of it are expressed:
(12)$$D_{h}C=0\;\;\;\;\left(\mathrm{here}\;C\;\mathrm{is}\;\mathrm{constant}\right),$$
(13)$$D_{h}x=\frac{(x+h)-x}{h}=1,$$
(14)$$D_{h}x^{n}=\sum_{k=0}^{n-1}\frac{n!}{k!(n-k)!}x^{k}h^{n-k-1}\;\;\;\;\left(n\in N\right),$$
(15)$$D_{h}\frac{1}{x}=-\frac{1}{x^{2}+hx},$$
(16)$$D_{h}\frac{1}{x^{n}}=\frac{1}{h{\left(x+h\right)}^{n}}-\frac{1}{hx^{n}},$$
(17)$$D_{h}e^{bx}=\frac{e^{bh}-1}{h}e^{bx}\;\;\;\;\left(\mathrm{here}\;b\in R\right),$$
(18)$$D_{h}a^{bx}=\frac{a^{bh}-1}{h}a^{bx}\;\;\;\;\left(\mathrm{here}\;b\in R\right).$$

In Figure 1, we illustrate the behavior of $D_{h}e^{x}$ at different values of *h* as an example. We could see that it behaves as an exponential when h=0. For any fixed values of *h*, $D_{h}e^{x}$ is a monotonically increasing function with respect to the variable *x*. The values of this derivative also increase when the parameter *h* becomes larger.

With the definition of *h*-derivative, V. Kac and P. Cheung [19] developed a type of quantum calculus, known as *h*-calculus. As a matter of fact, an operator such as *h*-derivative is called the forward difference quotient operator. Analogously, it also has the backward difference quotient operator $\nabla_{h}$ and the central difference quotient operator $\delta_{h}$, defined as [20]
(19)$$\nabla_{h}f\left(x\right)=\frac{f(x)-f(x-h)}{h},$$
(20)$$\delta_{h}f\left(x\right)=\frac{f\left(x+\frac{1}{2}h\right)-f\left(x-\frac{1}{2}h\right)}{h}.$$Note that the regular vector differential operator ∇ has been generalized based on *h*-derivative. We then explore the connection between the *h*-derivative entropy and its modified forms.

## 3. *h*-Derivative and Non-Additive Entropy

In order to generalize the non-additive entropic forms, one could utilize Equation (Equation 7) and give out the following equation:(21)$$S_{h}=-D_{h}\sum_{i=1}^{W}p_{i}^{x}{|}_{x=1}=-\frac{\sum_{i=1}^{W}p_{i}^{1+h}-1}{h},$$
with the normalization condition $\sum_{i=1}^{W}p_{i}=1$. When h→0, it will go back to the usual BG entropy. Note that it also recovers the Tsallis non-extensive entropy, $S_{q}$, cf. Equation (Equation 1) under the transformation of h=q−1.

Following the ways of the central difference quotient operator of Equation (Equation 20), we define a new form of two-parameter $\left(h,h^{\prime }\right)$-derivative,
(22)$$D_{h,h^{\prime }}f\left(x\right)=\frac{f\left(x+h\right)-f\left(x-h^{\prime }\right)}{h+h^{\prime }}\;\;\;\;\left(h,h^{\prime }\in R\right).$$The corresponding $\left(h,h^{\prime }\right)$-entropy is
(23)$$S_{h,h^{\prime }}=-D_{h,h^{\prime }}\sum_{i=1}^{W}p_{i}^{x}{|}_{x=1}=-\sum_{i=1}^{W}\frac{p_{i}^{1+h}-p_{i}^{1-h^{\prime }}}{h+h^{\prime }}.$$

Similarly, when $h=h^{\prime }\to 0$, the entropy $S_{h,h^{\prime }}$ returns to the BG one. It is shown that,
(24)$$\begin{array}{ccc} \lim_{h=h^{\prime }\to 0}S_{h,h^{\prime }} & = & -\lim_{h=h^{\prime }\to 0}\sum_{i=1}^{W}\frac{p_{i}^{1+h}-p_{i}^{1-h^{\prime }}}{h+h^{\prime }}\\ & = & -\lim_{h=h^{\prime }\to 0}\sum_{i=1}^{W}p_{i}\frac{e^{h\ln p_{i}}-e^{-h^{\prime }\ln p_{i}}}{h+h^{\prime }}\\ & = & -\lim_{h=h^{\prime }\to 0}\sum_{i=1}^{W}p_{i}\frac{e^{h\ln p_{i}}\ln p_{i}+e^{-h^{\prime }\ln p_{i}}\ln p_{i}}{2}\\ & = & -\sum_{i=1}^{W}p_{i}\ln p_{i}=S_{BG}. \end{array}$$Note that L’Hospital’s rule has been applied within the formula $de^{\alpha x}/dx=\alpha e^{\alpha x}$ for the last step in the above.

It is constructive to explore the connections with the already known statistical distributions. For example, the Tsallis entropy is obtained by $h=q-1,h^{\prime }=0$. While taking $h=q-1,h^{\prime }=1-q^{-1}$, we can obtain the Abe entropy Equation (Equation 38) (see the discussion in the Appendix A) [17]. The non-extensive entropy given in Equation (Equation 40) proposed by Borges and Roditi [18] (also see the Appendix A) can then be recovered with the relationship of $h=q-1,h^{\prime }=1-q^{\prime }$. Although the non-extensive entropy of Borges and Roditi and our two-parameter $\left(h,h^{\prime }\right)$-entropy have similar forms, we gained them using different mathematical methods. Specifically, we used $\left(h,h^{\prime }\right)$-derivative developed by ourselves, which differs from the *q*-calculus used by Borges and Roditi. In addition to the difference in the form of expression between the two-type derivative, a conspicuous point is that our two-parameter $\left(h,h^{\prime }\right)$-derivative does not require the function f(x) to be continuous and differentiable at x=0.

By assuming $h^{\prime }=h$, we could also obtain another new form of entropy $S_{h,h}$, which is obviously invariant under the interchange h↔−h. As a matter of fact, it is nothing new but the well-known Kaniadakis non-extensive κ-entropy [15]. Last but not least, it is set that $h^{\prime }\to -h$ and $h^{\prime }=-h+\delta$. Considering the limit of δ→0 and $h^{\prime }\to -h$, we could also cover the exact Shafee entropy [16,21] by taking the transformation of q=h+1.

In Table 1, we summarize different entropy functions, which can be represented by this two-parameter Sh,h′ entropy through taking different values of *h* and h′. In addition, by choosing h′=−1/h the function Sh,h′ becomes
(25)Sh,−1/h=−∑i=1Wpi1+h−pi1−1/hh+1/h,Note that this entropic form looks much similar to Abe entropy [17], but it is totally different in fact that Abe entropy cannot be recovered only by exchanging *q* and *h* when comparing them. Hereby, we name it the modified-Abe entropy function. Except for the entropy forms listed in Table 1, there is a well-known entropy—Renyi entropy, which can be related to Sh,h′ through the relationship between Renyi entropy and Tsallis entropy (only for q≤1) [22],
(26)SqRenyi≡ln∑i=1Wpiq1−q=ln1+(1−q)SqTsallis1−q.entropy-25-00918-t001_Table 1Table 1The two-parameter entropy Sh,h′ recovers other entropy functions by the variation of h,h′.Entropy TypeSh,h′Boltzmann–Gibbsh=h′→0Tsallis [5]h=q−1,h′=0or h=0,h′=1−qκ [15]h′=h=κ(κ,r) [23]h=r+κ,h′=κ−rγ [23]h=2γ,h′=γAbe [17]h=q−1,h′=1−q−1or h=q−1−1,h′=1−qShafee [16,21,24]h′→−hmodified Abeh′=−1/h

## 4. Properties

Now we shall address some properties of this $\left(h,h^{\prime }\right)$-entropy, $S_{h,h^{\prime }}$. As we all know, the Boltzmann–Gibbs and the Tsallis entropy can be expressed as [6,25]
(27)$$S_{BG}=-\langle \ln p_{i}\rangle =\langle \ln\left(1/p_{i}\right)\rangle ,\;S_{q}=\langle \ln_{q}\left(1/p_{i}\right)\rangle ,$$
where $\langle \ldots \rangle \equiv \sum_{i=1}^{W}p_{i}\left(\ldots \right)$ is the standard mean value, and $\ln_{q}$ is *q*-logarithm. Along this line, we straightforwardly obtain
(28)$$S_{h,h^{\prime }}=\langle \ln_{h,h^{\prime }}\left(1/p_{i}\right)\rangle ,$$
where $\ln_{h,h^{\prime }}$ is the $\left(h,h^{\prime }\right)$-logarithm, and it can be expressed as
(29)$$\ln_{h,h^{\prime }}\left(x\right)=\frac{x^{h^{\prime }}-x^{-h}}{h+h^{\prime }}.$$

### 4.1. Non-Negativity

First of all, we consider a thermal system within any possible state. The probability distribution of each microstate *i* is defined as $p_{i}$. If we assume $p_{i}^{1+h}⩾p_{i}^{1-h^{\prime }}$, namely, $1+h⩽1-h^{\prime }$, for $0⩽p_{i}⩽1$, thus can we obtain $h+h^{\prime }⩽0$ and this two-parameter entropy $S_{h,h^{\prime }}⩾0$.

### 4.2. Extremal at Equal Probabilities

Utilizing the Tsallis entropy, $S_{q}^{T}=\frac{\sum_{i=1}^{W}p_{i}^{q}-1}{1-q}$, this two-parameter entropy $S_{h,h^{\prime }}$ can be expressed with it as,
(30)$$S_{h,h^{\prime }}=\frac{1}{h+h^{\prime }}\left(hS_{1+h}^{T}+h^{\prime }S_{1-h^{\prime }}^{T}\right).$$For the Tsallis entropies inside this formula, namely $S_{1+h}^{T}$ and $S_{1-h^{\prime }}^{T}$, it is easy to know that both of them reach their extreme values when all the probabilities are equal [6]. Therefore, at the state of equal probability, our entropy $S_{h,h^{\prime }}$ will also approach to its extreme value since
(31)$$\frac{\;d}{\;dp_{i}}S_{h,h^{\prime }}=\frac{1}{h+h^{\prime }}\left[h\frac{\;d}{\;dp_{i}}S_{1+h}^{T}+h^{\prime }\frac{\;d}{\;dp_{i}}S_{1-h^{\prime }}^{T}\right]=0.$$

### 4.3. Expansibility

It is straightforwardly verified that $S_{h,h^{\prime }}$ is expansible for any values of *h* and $h^{\prime }$, since
(32)$$S_{h,h^{\prime }}\left(p_{1},p_{2},\cdots ,p_{W},0\right)=S_{h,h^{\prime }}\left(p_{1},p_{2},\cdots ,p_{W}\right).$$This property trivially follows from the definition itself. It means when we add some events with zero probabilities, $S_{h,h^{\prime }}$ keeps invariant.

### 4.4. Non-Additivity

When we consider a system that can be decomposed into two independent sub-systems, *A* and *B*, ($p_{ij}^{A+B}=p_{i}^{A}p_{j}^{B}$),
(33)$$\begin{array}{ccc} S_{h,h^{\prime }}\left(A+B\right) & = & -\sum_{i=1}^{W_{A}}\sum_{j=1}^{W_{B}}\frac{{\left(p_{ij}^{A+B}\right)}^{1+h}-{\left(p_{ij}^{A+B}\right)}^{1-h^{\prime }}}{h+h^{\prime }}\\ & = & S_{h,h^{\prime }}\left(A\right)\cdot \sum_{j=1}^{W_{B}}{\left(p_{j}^{B}\right)}^{1+h}+S_{h,h^{\prime }}\left(B\right)\cdot \sum_{i=1}^{W_{A}}{\left(p_{i}^{A}\right)}^{1-h^{\prime }}. \end{array}$$The values of *h* and $h^{\prime }$ cannot be zero together in case (or h=−1 and $h^{\prime }=1$ appear at the same time). In other words, $S_{h,h^{\prime }}$ is said to be non-additive similar to the Tsallis non-extensive entropy.

## 5. Summary and Outlook

To summarize, with the generalized *h*-derivative we firstly propose a two-parameter non-additive entropy, $S_{h,h^{\prime }}$, in order to connect several non-extensive entropy functions. The *h*-derivative motivated non-additive entropy, $S_{h,h^{\prime }}$, is demonstrated to recover different kinds of non-extensive entropy formalisms, such as the Tsallis entropy ($h=q-1,h^{\prime }=0$), the Abe entropy ($h=q-1,h^{\prime }=1-q^{-1}$), the Borges–Roditi entropy ($h=q-1,h^{\prime }=1-q^{\prime }$), the Kaniadakis κ-entropy ($h^{\prime }=h=\kappa$) and the Shafee ($h^{\prime }\to -h$ or $h^{\prime }=-h+\delta$, here δ→0) non-extensive entropy by varying values of $h,h^{\prime }$. On the other hand, the present two-parameter entropy exhibits all the relative properties as a generalized non-extensive entropy. Furthermore, the remarkable relationship between $S_{h,h^{\prime }}$ and other non-extensive entropies may cast a light on the connection of non-extensive entropy and some mathematical structures such as quantum calculus. It may lead to a deeper understanding of the mathematical and physical foundations of non-extensive statistics. We also noticed some other two-parameter distribution functions, such as the (r,q)-distribution and (α,κ)-distribution [26,27], which have been well-applied to astrophysics or space plasma physics. These two-parameter distributions provide another view to investigate the non-Maxwellian systems. It will be of great interest to associate this $\left(h,h^{\prime }\right)$-entropy with them and find out the deeper connections. There are also various elegant forms of entropy, such as fractional entropy [28] and Deng entropy [29]. Our $\left(h,h^{\prime }\right)$-entropy, $S_{h,h^{\prime }}$, is indeed unable to establish a connection with theirs. Further exploration of the inherent connections between different forms of entropy is necessary.

## Figures and Tables

**Figure 1 entropy-25-00918-f001:**
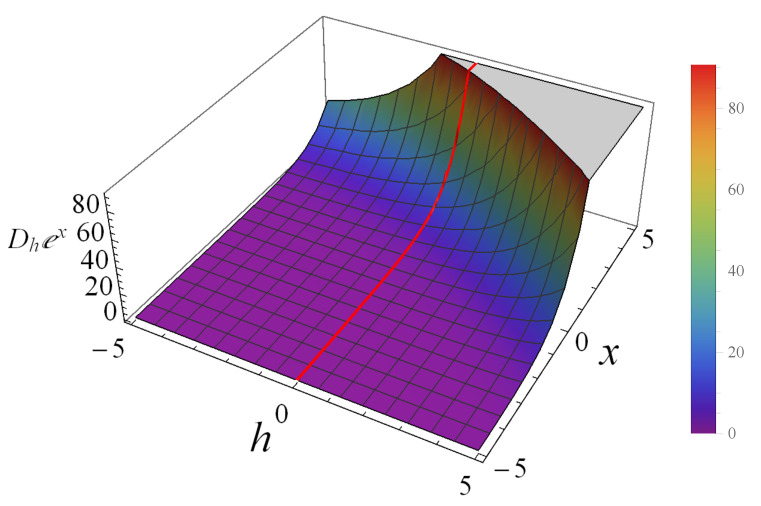
The behavior of $D_{h}e^{x}$ when *h* varying from −5 to 5. The red line denotes ${\left(e^{x}\right)}^{\prime }$.

## Data Availability

Not applicable.

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
