# Peer review of "A Note on the Connection between Non-Additive Entropy and h-Derivative"

_entropy, 2023, doi:10.3390/e25060918_

Round 1
Reviewer 1 Report
This paper is clearly written and appears to be mathematically correct. It introduces a two-parameter Jackson-like derivative and goes from there to obtain S_{h,h’}. It is nevertheless hard to recommend publication unless the authors provide a satisfactory answer to the following point: The entropy S_{h,h’} given in Eq. (22) reproduces that of Borges-Roditi in Physics Letters A 1998, 246, 399–402. Consequently, the authors need to explain/justify what is exactly their own original contribution in this paper.
Other points follow:
1) In the title and elsewhere, to avoid confusion in the literature, it should be “non-additive entropy” or “non-additive entropic functional” instead of “non-extensive entropy”.
2) In the Abstract: There are over 50 entropic functionals of all kinds in the literature. Therefore, “the major part” is wishful thinking. Something like “a wide part” or “a sensible part” is certainly more realistic.
3) After Eq. (24) the statement “but it is totally different in fact” should be explained/clarified.
Reviewer 2 Report
This article derives a new 2-parameterized non-extensive entropy using the h-derivative and establishes some connections between it and other non-extensive entropies. This is a very interesting result, but the authors are expected to address the following three issues.
The first is Renyi entropy and fractional entropy, which are also two well-known non-extensive entropies, can be related to the proposed unified formula?
The second is that the contribution of this paper is to give a new entropy unified formula based on h-derivatives, but the title does not reflect the existence of this unified formula.
I am also very interested about the work with Deng entropy published as Uncertainty Measure in Evidence Theory". Is it possible to derive Deng entropy or RPS entropy published as Entropy of Random Permutation Set?
Finally, the authors unify many non-extensive entropies with two parameters, h and h', which is interesting, but some explanations of the parameters in the corresponding entropies are unclear, can the authors qualitatively give the role of h and h', or can they verify h and h 's effect through more simulations?
In short, my opinion is that this work is very interesting and deserves the possible publication after revisions of the above minor issues.
Reviewer 3 Report
Dear Editor,
The paper "A note on the connection between non-extensive entropy and h-derivative" by Jin-Wen Kang et al. discusses the central part of entropy measures. They introduce a two-parameter non-extensive entropic form for the h-derivative, which generalizes the conventional Newton-Leibniz calculus. They provide a new entropy function to describe the non-extensive systems and recover several types of well-known non-extensive entropic expressions.
Related to the current manuscript, I have the following points.
Minor points:
1) I detect some paragraphs similar to Ref.[13].
2) In lines 30 and 31, Ref [13] must be cited. Ref.[6] has to be removed, an enormous list of papers that makes it difficult to assess their relevance to the manuscript. I suggest the authors give only references pertinent to the article.
Major point:
The word "non-extensive" is repeatedly used in the document. But it needs to be clarified the concept for this reviewer. An interesting discussion about "Non-additivity" is included in Section 4.4. A similar analysis needs to be made with "Non-extensivity," considering that systems can lose the additivity preserving the extensivity. The concepts are close but are different. This fact has to be discussed in the paper and conclusions.
This manuscript deserves to be published in Entropy after implementing the current suggestions.
Reviewer 4 Report
This manuscript considers nonextensive entropy that contains the existing ones in special cases and gives some relevant properties in the literature. The novelty is a generalization of the h-derivative defined in Eq. (21). Based on this, the authors introduce the two-parameter generalized entropy Eq. (22).
The calculations are elementary and the properties given are along the previous studies.
Thus, the title of the paper may contain the term of the generalized h-derivative to reflect the contents presented. Most of Section 2 is devoted to the known (or review) fact and seems not necessary, including Fig. 1. Also, the presentation has some defects.
If the authors consider the following points, it might become possible for acceptance for publication.
Below line 25 in Page 1, the readers can not know what the symbols in the formula denote.
The authors must delete Section 4.5 as this property is neither related to the stability nor validate any generalized entropy. It is rather related to the concept of the modulus of continuity: one of the basic features of uniformly continuous functions (See, textbooks of analysis).
There have been misunderstandings in a small group of the literature of Tsallis entropy.
Some other detailed concerns:
- To obtain Eq. (6), Eq. (5) is not necessary here.
- The prime symbol for the index h is somewhat confusing with the ordinary derivative in Eq. (29) and below.
- Below Eq. (29): it is unclear how f'=0=g' leads to the equiprobability x=1/W. Moreover, the relation h'f'+h'g'=0 does not mean f'=0=g'. It just means the constant ratio for f'/g'.
Several English grammatical errors exist in the text.
Several English grammatical errors exist in the text.
Round 2
Reviewer 1 Report
The present revised version is, in my opinion, agreeable for publication.
Author Response
Thanks deeply for the referee again.
Reviewer 3 Report
Dear Editor,
The minor points are partially solved. The removal of the reference [6] is Ok. The corresponding change of the word non-extensivity to non-additivity is clever and partially solves the problem because the expression non-extensive remains in the document. Tsallis entropy is a non-additive measure, but not necessarily non-extensive, but this subject is understood by the authors. However, a new conceptual problem emerges in the comment after Eq.(2) in the last version. For decades, discussing interpreting the parameter T in all generalized frameworks has been an open issue beyond the Tsallis entropy. My suggestion is to replace a sentence as follows:
"T denotes the temperature of system" ==> "T, a variational parameter representing the temperature when the system reaches equilibrium,"
After solving this suggestion, the manuscript deserves to be published.
Author Response
Thank the reviewer's nice comments and in the new manuscript we have replaced the corresponding sentence.
Reviewer 4 Report
This revision is unsatisfactory for acceptance.
Some flaws remain in the presentation.
Instead of removing, leaving the text of 4.5 as Appendix B is definitely inappropriate. This part adds nothing to stability and is a red herring for all readers.
Response 4: I do not see any change.
Response 5: How do f'=0 and g'=0 lead to x=1/W?
The authors added Eq. (26) and state that Renyi entropy can be related to S_h,h', but they do not provide the relation below. Why did you add this part?
Many errors remain in the text.
Round 3
Reviewer 4 Report
The authors revised the manuscript rightly, and it can be a publishable form.
The text needs the editing for correctness.